# PPAR-Gamma Activation May Inhibit the In Vivo Degeneration of Bioprosthetic Aortic and Aortic Valve Grafts under Diabetic Conditions

**DOI:** 10.3390/ijms222011081

**Published:** 2021-10-14

**Authors:** Shintaro Katahira, Yukiharu Sugimura, Sophia Grupp, Robin Doepp, Jessica Isabel Selig, Mareike Barth, Artur Lichtenberg, Payam Akhyari

**Affiliations:** Department of Cardiovascular Surgery, Medical Faculty and University Hospital Düsseldorf, Heinrich-Heine-University Düsseldorf, 40225 Düsseldorf, Germany; shinkatahira@med.tohoku.ac.jp (S.K.); Yukiharu.Sugimura@med.uni-duesseldorf.de (Y.S.); Sophia.Grupp@med.uni-duesseldorf.de (S.G.); Robin.Doepp@med.uni-duesseldorf.de (R.D.); Jessica.Selig@med.uni-duesseldorf.de (J.I.S.); Mareike.Barth@med.uni-duesseldorf.de (M.B.); Payam.Akhyari@med.uni-duesseldorf.de (P.A.)

**Keywords:** diabetes mellitus, allograft, bioprosthetic valve, degeneration, PPAR-gamma, pioglitazone

## Abstract

Background: We aimed to examine the anti-calcification and anti-inflammatory effects of pioglitazone as a PPAR-gamma agonist on bioprosthetic-valve-bearing aortic grafts in a rat model of diabetes mellitus (DM). Methods: DM was induced in male Wistar rats by high-fat diet with an intraperitoneal streptozotocin (STZ) injection. The experimental group received additional pioglitazone, and controls received normal chow without STZ (*n* = 20 each group). Cryopreserved aortic donor grafts including the aortic valve were analyzed after 4 weeks and 12 weeks in vivo for analysis of calcific bioprosthetic degeneration. Results: DM led to a significant media proliferation at 4 weeks. The additional administration of pioglitazone significantly increased circulating adiponectin levels and significantly reduced media thickness at 4 and 12 weeks, respectively (*p* = 0.0002 and *p* = 0.0107, respectively). Graft media calcification was highly significantly inhibited by pioglitazone after 12 weeks (*p* = 0.0079). Gene-expression analysis revealed a significant reduction in relevant chondro-osteogenic markers *osteopontin* and *RUNX-2* by pioglitazone at 4 weeks. Conclusions: Under diabetic conditions, pioglitazone leads to elevated circulating levels of adiponectin and to an inhibition of bioprosthetic graft degeneration, including lower expression of chondro-osteogenic genes, decreased media proliferation, and inhibited graft calcification in a small-animal model of DM.

## 1. Introduction

Recently, diabetes mellitus has been identified as a risk factor for the development of calcific aortic valve disease (CAVD) [1,2]. Real-world trends from the clinical field show an increasing volume of invasive procedures to treat cardiovascular complications of diabetes mellitus, including valvular heart disease, often requiring prosthetic valve replacement [3].

With respect to heart valve prostheses, biological solutions omitting lifelong anticoagulant therapy have been increasingly favored world-wide, despite durability limitations inherently associated with biological valvular prostheses. In this context, diabetes has also been reported as a risk factor for structural valve deterioration, which emphasizes the need for additive postoperative therapy measures to improve the durability of bioprosthetic valves, particularly in young patients with diabetes.

Pioglitazone, which is one of the thiazoline derivatives, has been applied clinically as a PPARγ agonist with antidiabetic effects by enhancing insulin sensitivity for enhanced blood glucose control in diabetes [4]. However, above the aforementioned antidiabetic effects, pioglitazone has also been reported to improve lipid metabolism, and further to trigger anti-atherogenic and anti-inflammatory effects by modulating the expression of inflammatory cytokines [5,6].

Previously, a translational model for chronic in vivo evaluation of function and durability of biological aortic valvular grafts has been reported to provide valuable reproducible results. Although there are reports that pioglitazone suppresses calcification of the native aortic valve, little is known on its effect on bioprosthetic valves and aortic grafts. By combining this model with a well-established model of diabetes, sought to evaluate the potentially beneficial effect of pioglitazone on calcific degeneration of bioprosthetic aortic and aortic valve grafts. Histological analysis, blood measurements, and gene-expression analyses were performed. This study intended to evaluate the potential of PPARγ activation for improving the chronic functional durability of bioprosthetic grafts in diabetic recipients undergoing implantation of aortic grafts.

## 2. Results

### 2.1. Body Weight, Blood Glucose Changes, and Plasma Analysis

Table 1 shows the blood glucose levels and results of plasma analysis at the start of the experiment and at the time of aortic graft explantation. Blood glucose levels were significantly higher in both groups with diabetic conditions (DM, DM+Pio) at both time points of 4 weeks and 12 weeks as compared to the start of the experiments. In addition, diabetic animals (DM and DM+Pio) showed significantly higher body weight when compared to controls, both at 4 weeks and 12 weeks, respectively. Plasma analysis revealed that total cholesterol, triglycerides, and HDL at 4 weeks were significantly higher in the DM group than in the control group. Pioglitazone administration led to significant reduction in total cholesterol. At 12 weeks, animals in the DM group showed significantly higher levels of HDL when compared to the control group.

Circulating levels of adiponectin were measured from all animals at the end of the experiments. The induction of DM did not change the circulating levels of adiponectin, and a change over time from 4 weeks to 12 weeks could not be determined (Figure 1). However, pioglitazone administration significantly increased the plasma levels of adiponectin at 4 weeks (Figure 1A) and at 12 weeks (Appendix A), respectively, when compared to diabetic animals and also in comparison to non-diabetic controls (under pioglitazone administration at both analyzed time points, *p* < 0.001 for both time points compared to non-diabetic animals, and *p* = 0.004 and *p* = 0.02 for comparison to diabetic animals at 4 and 12 weeks, respectively).

### 2.2. Histological Changes and Remodeling of the Aortic Wall

The tubular segments of the aortic grafts (segments A2, B1, B2, Figure 2) were analyzed for signs of remodeling and degeneration. Therefore, tissue specimens of four animals in each group and at each time point were used for histological evaluation. Systematic quantification was applied to determine neo-intima hyperplasia and media proliferation as key features of bioprosthetic graft degeneration (Figure 3). At the level of graft intima, pioglitazone led to a trend of lower thickness at 4 weeks (*p* = 0.0776) and comparable intima thickness at later time point (data not shown). Diabetic conditions led to a significant media thickening in the DM group already at 4 weeks as compared to the control group (*p* = 0.0002). Administration of pioglitazone inhibited the adverse impact of diabetes (group DM+Pio), resulting in significantly lower thickness of the bioprosthetic grafts when compared to the diabetic group without pioglitazone medication (*p* = 0.0149). At 12 weeks, the greatest values for media thickness were again observed in the diabetic group, however, a numeric increase in mean values of graft thickness was also observed in control animals. Administration of pioglitazone resulted in significantly smaller thickness values despite the presence of diabetic conditions (DM+Pio vs. DM, *p* = 0.0080).

Calcification of the bioprosthetic grafts in the tubular segments of the aortic grafts was next analyzed using von Kossa staining and a standardized scoring system (Figure 4). At the early time point of 4 weeks there was no significant difference in graft calcification (data not shown). However, at 12 weeks, there was a marked increase in calcification in bioprosthetic grafts of the DM group compared to the control group (*p* = 0.0534, Figure 4). Administration of pioglitazone significantly inhibited bioprosthetic graft calcification despite the presence of diabetic conditions (DM vs. DM+Pio, *p* = 0.0101).

To gain a better insight into the pattern of bioprosthetic graft deterioration, a second method was employed to quantify the level of regional distribution of biomineralization (Appendix A). The proportion of cross-sectional area staining positive for Alizarin red showed a similar order as already determined with software-based evaluation of von Kossa stainings. Under diabetic conditions, a marked increase in biomineralized cross-sectional area was observed, when compared to results of the control group or diabetic group with additional pioglitazone administration, respectively. However, due to the range of variation, statistical significance was not achieved in the applied sample size (Appendix A).

### 2.3. Degenerative Changes at the Level of the Bioprosthetic Aortic Valve

Calcification of the proximal segments of bioprosthetic grafts including the aortic valve prosthesis were analyzed with a sub-segmental focus on three anatomic sites: valvular leaflets, annulus, and commissural region, using von Kossa staining (Figure 5A). Analysis by the scoring system revealed that both the DM and DM+Pio groups had a significantly higher calcification score than the control group (DM vs. control, *p* = 0.0485; DM+Pio vs. control, *p* = 0.0485). There was no significant difference in the calcification score between the three groups in the annulus region. In the area of the commissures, the DM group showed a significantly higher calcification rate than the control group (DM vs. control, *p* = 0.0270), whereas additional pioglitazone treatment abolished this effect. Overall, analysis of all regions revealed significantly higher calcification scores in the DM (*p* = 0.0002) and the DM+Pio group (*p* = 0.0073) compared to the control group. Nevertheless, there were no significant differences between the DM and the DM+Pio group (Figure 5B). Results of Alizarin red staining were in line with these findings (Appendix A).

### 2.4. Gene-Expression Analysis of Explanted Bioprosthetic Grafts

In the analysis of inflammatory markers, no significant difference was found among the three groups for either time point of 4 or 12 weeks, respectively, with regard to *TNF-α*, *IL-1b*, *IL-6*, and *RAGE* (Appendix A). Nevertheless, beside genes involved in the inflammatory response, genes associated in chondro-osteogenic transformation were also analyzed. Here, pioglitazone led to a significant suppression of *bone morphogenic protein 2* (*p* = 0.0386) as well as *osteopontin* (*p* = 0.0074) at 4 weeks when compared to controls as well as to diabetic recipients. The expression of osteocalcin as well as the transcription factor *Runx2* remained comparable between the three different groups at both time points (Figure 6).

Other genes that play a role in cell differentiation (*ACTA2*) and tissue degeneration (*OPG*), as well as collagen remodeling (*MMP2*), were partially altered in a treatment- and time-dependent manner, showing a reduced level of expression under diabetic condition irrespective of pioglitazone treatment (Appendix A). Other markers related to inflammation (*NFκB, MMP9*) were, in contrast, not affected by any treatment, whereas *MCP1* expression was numerically reduced by diabetic conditions, particularly at 4 weeks when pioglitazone was added, however the latter findings were without statistical significance (Appendix A).

### 2.5. Identification of Cellular Remodeling

Immunostaining of cross-sections was performed using antibodies against αSMA/vWF at 12 weeks. Here, αSMA-positive cells were observed not only in the media but also in the intima of explants of all groups. In the control group, vWF was positive in the intima, and the normal vascular structure was maintained in most cases (Figure 7).

## 3. Discussion

The present work was conducted to evaluate the potential benefit of pharmacological PPARy activation on functional durability of biological aortic valved grafts in the setting of diabetic conditions. Although AV-bioprosthesis calcification and also calcification of aortic graft material can be investigated using this model, the respective biomechanics of course differ in the heterotopic position. As major findings, this study demonstrates that the daily oral administration of pioglitazone as a PPARy activator is capable of significantly inhibiting the calcific degeneration of a bioprosthetic aorta and aortic valve in diabetic rats. These protective effects on the morphological level were accompanied by favorable modulation of gene expression, with significantly lower levels of *osteopontin* and *bone morphogenic protein 2*.

Diabetes is known as a condition that promotes atherosclerosis and vascular calcification. Thus far, diabetes models have been created using mice and rats, and their pathophysiology and various pharmacological approaches for the treatment of diabetes have been reported [7,8,9].

Obese Zucker rats and Otsuka Long–Evans Tokushima fatty rats are known as type 2 diabetes models with obesity [10,11]. We thought that more clinical data could be obtained by using a rat model of so-called ‘obese type 2 diabetes rats’, which more closely reflects the common scenario observed clinically. Therefore, we used a type 2 diabetes model that was based on two different interventions: a low dose of STZ injected intraperitoneally and leading to a relative insulin deficiency in combination with a high-cholesterol diet, which has been reported by Mansor et al. [8]. Compared with the control group, the body weight of both the DM group and the DM+Pio group increased significantly, and the DM group showed an increase in triglycerides, which is indicative of an obese rat.

Pioglitazone is a pharmacological compound that has been clinically used as a therapeutic drug against diabetes [12,13]. After clinical use, various clinical studies have shown a protective effect on the cardiovascular system. In the PRO active study, pioglitazone showed an inhibitory effect on the onset of cardiovascular events in diabetic patients with a history of macroangiopathy [14]. Furthermore, the CHICAGO study showed that pioglitazone suppresses the progression of arteriosclerosis in diabetic patients, which in turn enables primary prevention of cardiovascular events [15]. It was suggested that pioglitazone is excellent in maintaining good glycemic control for a long period of time and also leads to suppression of the onset of microangiopathy. However, it is currently unknown whether PPARy agonists may have a protective effect on cardiovascular bioprosthetic grafts in diabetic recipients. 

Vascular calcification can be broadly divided into two types: atherosclerotic calcification, which is calcification of intima-like atherosclerosis, and Menkeberg-type calcification, which is calcification of the media of blood vessels [16]. In the former, calcification occurs inside the so-called plaque in the process of plaque formation and its development. In the latter, vascular smooth muscle cells existing in the media are considered to have changed to osteoblast-like cells, causing so-called ossification inside the blood vessels. In the current work, features of both fundamental types of calcification were observed: smooth muscle cells and biomineralization were observed in the media and atherosclerosis was observed in the intima. In addition, vascular smooth muscle cells are also considered to be involved in intima calcification. Both calcification types are of particular concern in the context of diabetes mellitus. Although no significant difference was observed in intima calcification in this study, medial calcification was markedly present and as such significantly suppressed in diabetic animals treated with pioglitazone. Here, we observed significantly lower media calcification levels as well as significantly lower gene-expression levels for two crucial markers of chondro-osteogenic transformation, i.e., *osteopontin* and *bone morphogenic protein*, when pioglitazone was administered to diabetic aortic valve recipients. *BMP2* and *OPN* are produced during bone formation (ectopic calcification). In this experiment, the high value in the DM group at 4 weeks was considered to be indicative of activated bone formation. We interpret these findings as suggestive of an increased activation of remodeling at the earlier time point and less activation at 12 W. Moreover, other markers of tissue remodeling, i.e., *MMP2* and *9*, as well as alpha smooth muscle actin, were modified in their temporal gene expression during the chronic postoperative period in the DM+Pio group, which may explain the differences in media proliferation and calcification as a result. We speculate that pioglitazone may have acted as an inhibitor on the pathological transformation of vascular smooth muscle cells to osteoblast-like cells. However, our RT-PCR results could not identify inflammatory processes as a possible mechanism by which pioglitazone may have interfered with medial calcification. In future work, additional experiments focusing on other organ systems as possible intermediate targets for the action of PPARy agonists may aid in identification of further mechanisms underlying the protective effects of pioglitazone on bioprosthetic aortic valves under diabetic conditions.

The pathophysiology underlying calcific (native) aortic valve disease on the one hand and bioprosthetic-valve-deterioration calcification on the other is complex and not yet entirely understood. However, it is thought that osteoblast-like cells play a pivotal role in both processes [15,16,17]. The applied in vivo model included the induction of native valve insufficiency by wire puncture in the recipient animal. Due to the pathological level of shear stress induced by experimental AR, the native aortic valve could not be evaluated correctly. However, when the native descending thoracic aorta was analyzed, there was no difference in calcification between diabetic animals, irrespective of a treatment with pioglitazone. As the applied model did not result in a severely diabetic condition, we interpret these results as a confirmation of the rather mild alteration of the systemic susceptibility to calcific degeneration. There may be no change in the recipient’s aorta, however, in the presence of additional stimuli, e.g., local milieu of cryopreserved bioprosthetic grafts, and treatment-dependent differences in graft degeneration may be observed. 

Biomechanical abnormalities such as hypertension, increased cusp-tissue extension, and increased shear stress, increased oxidative stress, inflammation, hyperactivation of the renin-angiotensin system, and abnormal mineral metabolism, are thought to be involved in valve calcification [18,19]. Type 2 diabetes has been reported to cause chronic inflammation and increased AGE production, which are thought to promote calcification [20]. RT-PCR did not show significant differences regarding the inflammation markers *RAGE*, *TNFα*, and *IL-6*, but calcification was enhanced in the DM group, and factors other than those measured along this work most likely are involved in valve calcification. 

As one of the important anti-arteriosclerotic effects of pioglitazone, an increase in circulating adiponectin concentrations has been described, which in turn has been reported to exert vascular protective effects [21]. Our results on significantly elevated levels of circulating adiponectin induced by pioglitazone suggest that this mechanism is indeed a relevant effect of this pharmacological intervention and may be partly responsible for the protective effects that we observe in the bioprosthetic aortic implants. On the other hand, adiponectin levels have been measured in plasma and might therefore not necessarily correlate with the temporal expression levels of the analyzed degenerative markers in the graft tissue. As blood glucose levels did not significantly differ between the 4 and 12 week time-points within the DM+Pio groups (Table 1), consistently high adiponectin plasma levels over time here might rather delineate the stable effect of pioglitazone treatment. However, altered-tissue gene expression of chondro-osteogenic markers may be indicative of an early strong response at 4 weeks versus a rather masked effect of pioglitazone (and subsequently of adiponectin) at 12 weeks. The latter effect may be due to a generally higher pro-osteogenic state of the tissue as a result of the strong induction of diabetes and its subsequent tissue damage at 12 weeks. Future work will focus on the downstream signaling effects, possibly revealing further targets for a more specific therapeutic approach.

The results of this study have to be regarded in the context of several limitations. First, the severity of the induced diabetes in the rat model could not be kept constant. Second, since pioglitazone was applied as a component into the specific diet, there is the possibility that the exact daily dose of pioglitazone may have varied among the individual animals. However, the amount of chow intake as well as body weight development as a further measure of total chow intake revealed an increase in both parameters, suggesting a constant administration of pioglitazone in compliance with the experimental plan. 

At the late time point of 12 weeks, our readout of media thickness delivered the lowest values for the group of DM+Pio. This was indeed unexpected. In view of the available results, one may speculate whether pioglitazone may exert effects on the remodeling process that are independent of the presence or absence of diabetic conditions. For more in-depth investigation on this particular question, an additional experimental group of animals without DM but receiving pioglitazone may have been helpful. However, the focus of this study has been put on elucidation of a difference between DM + P and DM groups. 

It is possible that the surgical trauma and the respective postoperative inflammatory response may have varied in the very early phase, i.e., postoperative day 1 to 10. Moreover, due to the limited aortic valve tissue available from each animal, histological analyses and PCR experiments were performed on explants derived from different rats. Hence, histological results are not necessarily identical to the results of gene-expression analysis.

## 4. Materials and Methods

### 4.1. Animals and Experiment Protocol

The experimental protocol is shown in Figure 8. A total of 60Wistar rats (250–300 g) were divided into three groups, a control group (Control; 4W *n* = 10, 12W *n* = 10), a diabetes mellitus group (DM; 4W *n* = 10, 12W *n* = 10), and a diabetes mellitus group receiving pioglitazone (DM+Pio; 4W *n* = 10, 12W *n* = 10). DM was achieved as previously described. Preparation of cryopreserved donor grafts [22], induction of regurgitation of recipient native aortic valve [23], and heterotopic bioprosthetic aortic valve implantation were performed in a standardized fashion as previously reported (Appendix A). Four weeks after implantation procedure, 10 animals in each group were sacrificed and aortic valved grafts were collected. The remaining 10 rats from each group were similarly sacrificed 12 weeks later. Four grafts of each group and each time point were used for histological examination and six grafts were used for RT-PCR. Blood was collected from each animal by sampling the inferior vena cava at the time of sacrifice. Blood glucose levels were measured at the start of the high-fat diet, before injection of STZ, at the time of AR preparation, at the time of implantation of the bioprosthesis, and at explantation, by collecting blood from the tail vein with a blood glucose test meter. Blood glucose measurements were performed with a fasting period of 4 h preceding the blood sampling time point.

### 4.2. Donor Graft Harvesting and Cryopreserved Graft Preparation

A total of 60 Wistar rats were euthanized under general anesthesia and then thoracotomy was immediately performed for harvesting of valve-bearing aortic graft collected with surrounding tissue. The graft was trimmed under a microscope as shown in Appendix A. The coronary arteries ware ligated with 8-0 monofilament. The graft was immersed in a conservation medium (Dulbecco’s modified Eagle’s medium +10% dimethylsulfoxide +20% fetal calf serum at 5 °C) and stored at −80 °C. The valve-bearing aortic grafts were prepared 1–2 weeks before prior to transplantation [22].

### 4.3. Blood Serum Analyses

At the time of explantation at 4 weeks or 12 weeks, blood samples were collected from the inferior vena cava using a heparin-containing syringe and centrifuged (4 °C, 1000 rpm, and 15 min). Serum specimens were evaluated at the Institute of Clinical Chemistry and Laboratory Diagnostics, Medical Faculty and University Hospital, Heinrich-Heine-University Düsseldorf, Germany. The following parameters were quantified at the serum level: potassium, calcium, creatinine, total cholesterol, triglycerides, high-density lipoprotein (HDL), low-density lipoprotein (LDL), aspartate aminotransferase (AST), and alanine aminotransferase (ALT).

### 4.4. Histological Evaluation

The explanted graft of each animal was divided into four sections: A1 (aortic root and aortic valve), A2 (ascending aorta to proximal aortic arch), B1 (distal aortic arch), and B2 (descending aorta) (Figure 2). Each section was rapidly frozen at −80 °C. Histological sections were prepared and processed according to standard protocols as previously described (see Appendix A).

### 4.5. Gene-Expression Analysis

Changes in the gene expression of bioprostheses as induced by the diabetes model and further modified by pioglitazone medication were evaluated on an mRNA level according to standard protocols using the ∆∆ Ct and a StepOnePlus cycler (Applied Biosystems, Waltham, MA, USA) as previously described [22] (see Appendix A). A list of the used primers and respective sequences is provided in Appendix A.

### 4.6. Statistical Analysis

Statistical analysis was performed using GraphPad Prism version 8.0 (GraphPad Software, San Diego, CA, USA). Comparisons between three groups were compared with Kruskal–Wallis and Dunn’s post-hoc test and two-way ANOVA with Tukey’s post-hoc test. All data are expressed as means ± standard error of mean.

## 5. Conclusions

Cryopreserved aortic valved grafts transplanted into a rat diabetes model showed calcification and arteriosclerotic lesions. Administration of pioglitazone led to significantly increased circulating adiponectin levels. Furthermore, pioglitazone significantly inhibited bioprosthetic calcification and resulted in lower expression levels of genes associated with chondro-osteogenic calcification despite diabetic conditions. Pioglitazone may delay bioprosthetic-aortic-valve deterioration in diabetic recipients. Further work is needed for elucidation of the mechanisms underlying the protective effects of pioglitazone. 

## Figures and Tables

**Figure 1 ijms-22-11081-f001:**
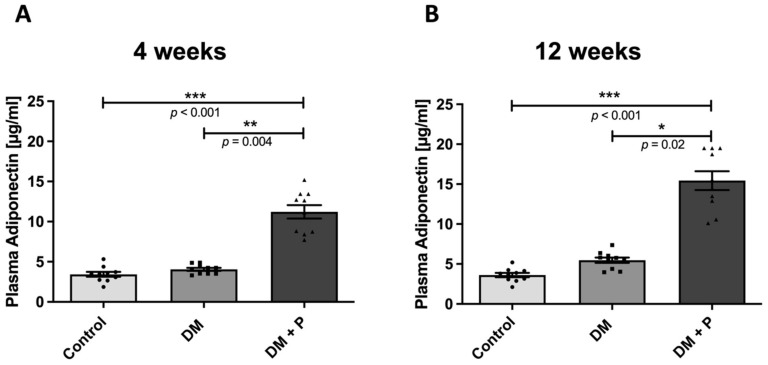
Circulating adiponectin levels. Plasma probes (of *n* = 10 animals per group) were analyzed by enzyme-linked immunosorbent assay (ELISA) for the level of circulating adiponectin at 4 weeks (**A**) and 12 weeks (**B**), respectively. Animals with diabetic conditions (DM) showed no significant change in circulating adiponectin levels when compared to non-diabetic controls (Control) at either time point. The addition of pioglitazone (DM+Pio) led to a significant threefold increase in plasma adiponectin levels at 4 weeks and at 12 weeks, respectively. DM, diabetes mellitus; Pio, pioglitazone; * *p* < 0.05; ** *p* < 0.01; *** *p* < 0.001. Data were analyzed using Kruskal–Wallis with Dunn’s multiple comparisons test.

**Figure 2 ijms-22-11081-f002:**
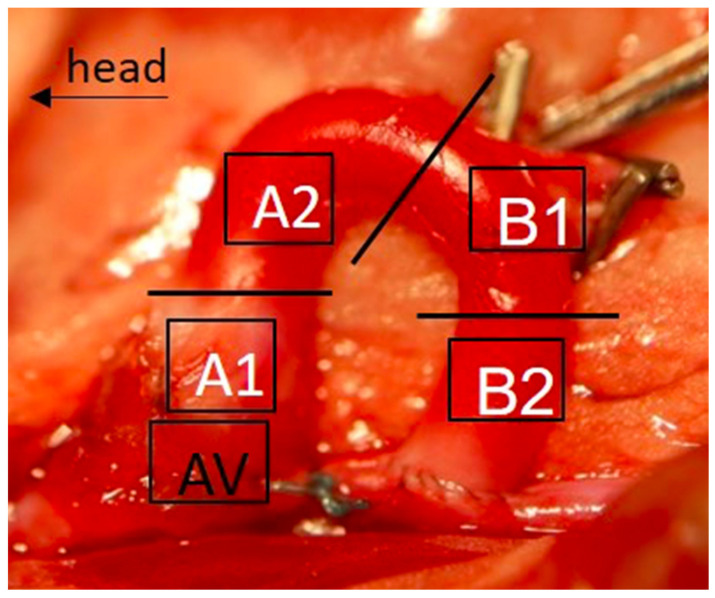
Aortic-valve-insufficiency operation, graft implantation. A graft was implanted into the recipient abdominal aorta. After explantation, the graft was divided into four parts, A1 (proximal ascending aorta and aortic valve), A2 (distal ascending aorta), B1 (proximal descending aorta), and B2 (distal descending aorta). AV, aortic valve.

**Figure 3 ijms-22-11081-f003:**
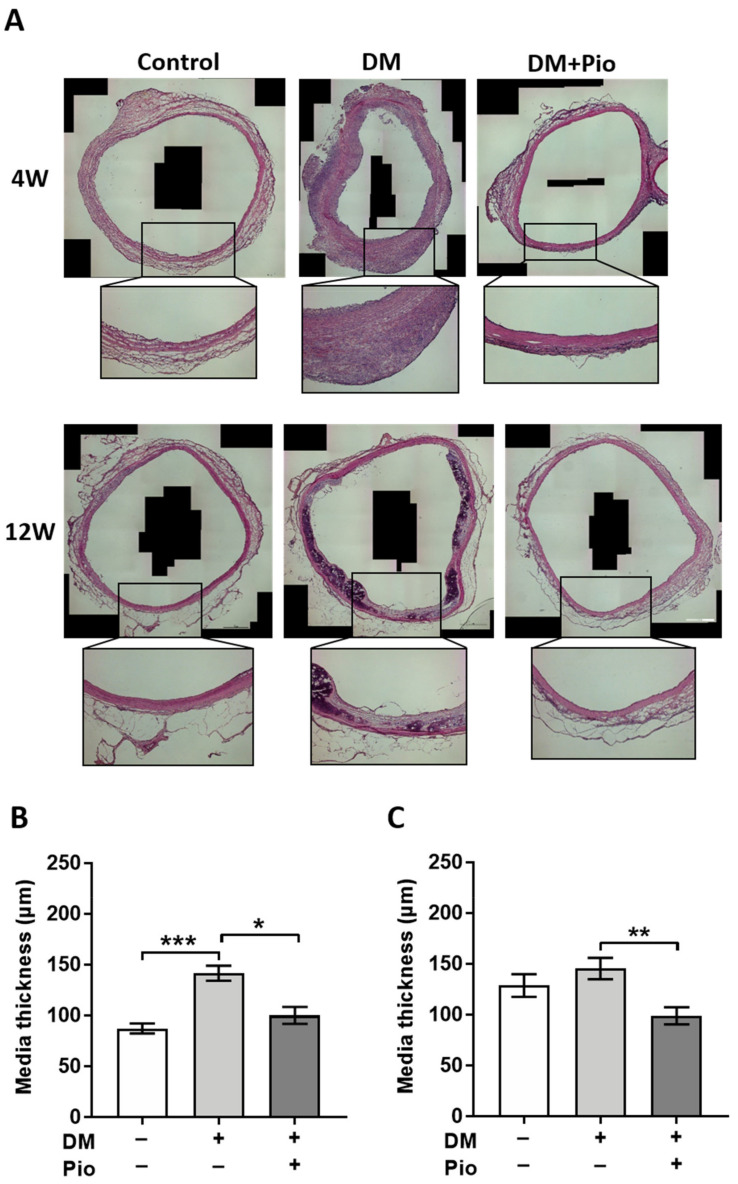
Histological results of aortic region—media thickness. For each group, tissue of *n* = 4 animals was stained and evaluated. (**A**) A representative hematoxylin eosin-stained image for each group, including higher-magnification inserts. After 4 weeks, DM had a significant thickening of the media, but it was not statistically different between the control and the pioglitazone-administered group. After 12 weeks, thickening of the media with calcification was observed in the DM group. (**B**) Measurement result of the media after 4 weeks. There was a significant difference in medial thickening in the DM group. (**C**) Even after 12 weeks, the medial thickening of the DM group was significantly different. DM, diabetes mellitus; W, weeks; Pio, pioglitazone; * *p* < 0.05; ** *p* < 0.01; *** *p* < 0.001. Data were analyzed using Kruskal–Wallis with Dunn’s multiple comparisons test.

**Figure 4 ijms-22-11081-f004:**
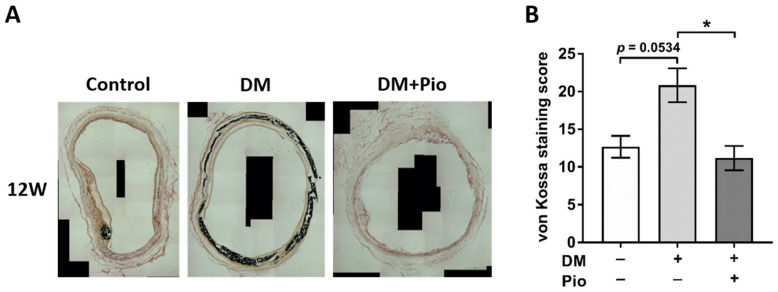
Histological results of aortic region—media calcification. For each group, tissue of *n* = 4 animals was stained and evaluated. (**A**) Representative von Kossa stainings after 12 weeks. The media was extensively calcified in the DM group. (**B**) Calcification scores after 12 weeks. Here, calcification was more advanced in the DM group. DM, diabetes mellitus; W, weeks; Pio, pioglitazone; * *p* < 0.05. Data were analyzed using Kruskal-Wallis with Dunn’s multiple comparisons test.

**Figure 5 ijms-22-11081-f005:**
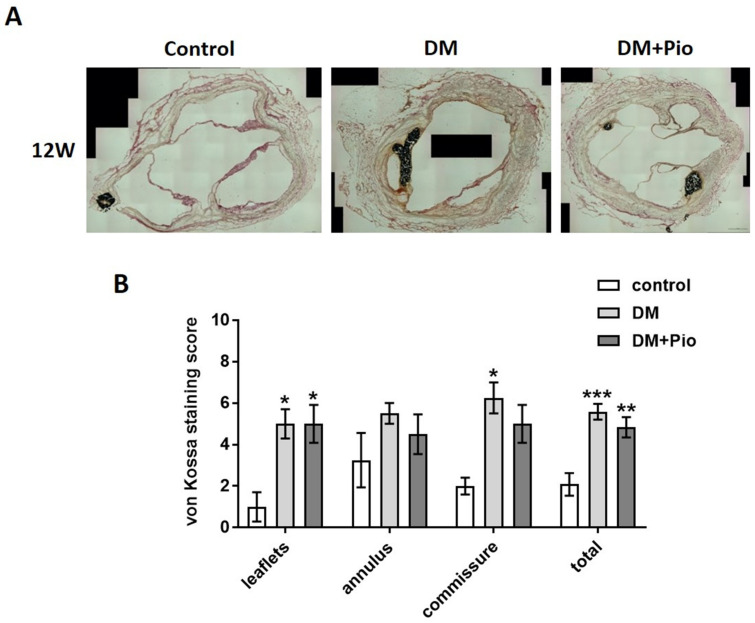
Histological results of aortic region –calcification of different regions. For each group, tissue of *n* = 4 animals was stained and evaluated. (**A**) Representative images of von Kossa staining of cross-sections at the level of the aortic valve after 12 weeks. Calcification of valve leaflets, annulus, and commissures was observed in all groups, with increased levels in DM and DM+Pio. (**B**) Semi-quantitative analysis for sub-segments of the aortic root as well as the entire cross-section. Highest calcification levels were observed in sub-segments of commissures. DM, diabetes mellitus; W, weeks; Pio, pioglitazone; * *p* < 0.05; ** *p* < 0.01; *** *p* < 0.001. Data were analyzed using two-way ANOVA with Tukey’s multiple comparisons test.

**Figure 6 ijms-22-11081-f006:**
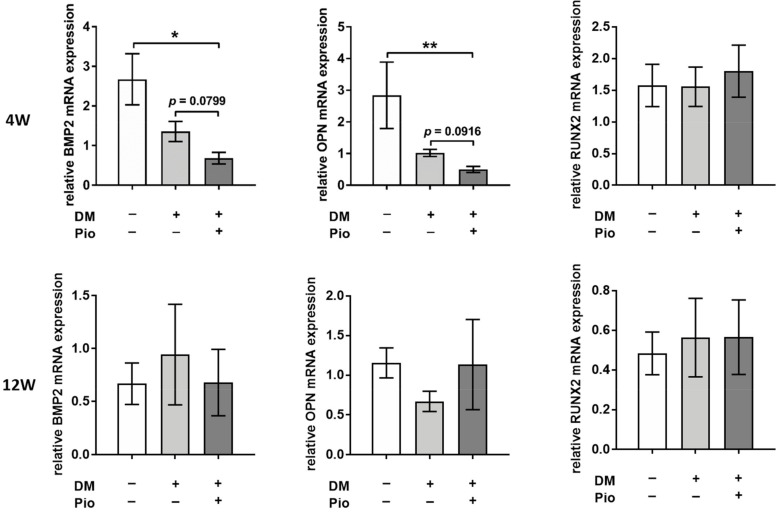
Gene-expression analysis of degenerative markers. Gene-expression analysis of aortic grafts after 4 and 12 weeks, respectively, revealed significant differences regarding key markers of calcific degeneration. For each group, tissue of *n* = 6 animals was analyzed. At the earlier time point of 4 weeks, bone morphogenic protein 2 and *osteopontin* expression were significantly reduced by pioglitazone (DM+Pio vs. DM and DM+Pio vs. control). This difference was lost at the later time point of 12 weeks. *BMP2*, Bone morphogenetic protein 2; *OPN*, *osteopontin*; *Runx2*, *Runt-related transcription factor 2*; * *p* < 0.05; ** *p* < 0.01. Data were analyzed using Kruskal–Wallis with Dunn’s multiple comparisons test.

**Figure 7 ijms-22-11081-f007:**
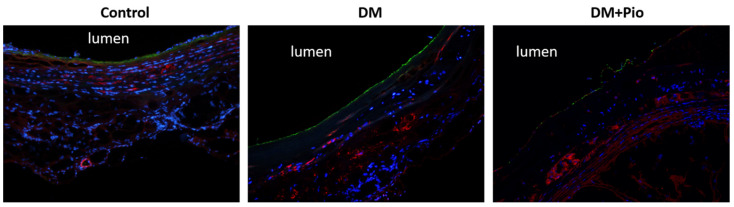
Immunohistology. For each group, tissue of *n* = 4 animals was stained and evaluated. Representative images of cross-sections subjected to immunostaining for αSMA and vWF after 12 weeks in vivo demonstrates a myofibroblast activation (αSMA+) in the media of diabetic groups (DM, DM+Pio). Endothelial cells (vWF+) were detected in all groups at the luminal surface. DM, diabetes mellitus; W, weeks; Pio, pioglitazone; αSMA, alpha smooth muscle actin; vWF, von Willebrand factor; DAPI, 4′,6-diamidino-2-phenylindole.

**Figure 8 ijms-22-11081-f008:**
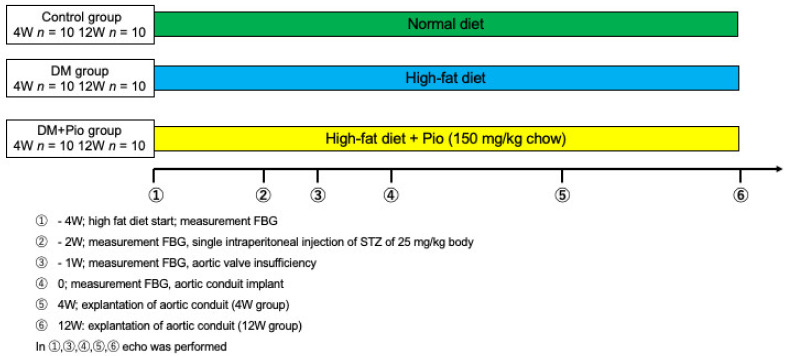
Experimental protocol. Recipient Wistar rats (*n* = 60) were divided into three groups. A high-fat diet was given for 2 weeks, after which STZ was injected intraperitoneally. Aortic valve insufficiency was created 1 week after injection and the graft was transplanted 1 week later. Animals were sacrificed 4 and 12 weeks after transplantation. DM, diabetes mellitus; W, weeks; FBG, fasting blood glucose; Pio, pioglitazone; STZ, streptozotocin.

**Table 1 ijms-22-11081-t001:** Body weight, blood glucose changes, and plasma results.

		4 Week			12 Week	
	Control	DM	DM + *p*	Control	DM	DM + *p*
	(*n* = 10)	(*n* = 10)	(*n* = 10)	(*n* = 10)	(*n* = 10)	(*n* = 10)
Body Weight(g)						
Start	274.8 ± 13.8	270.7 ± 19.8	273.5 ± 22.6	271.2 ± 11.3	263.2 ± 17.4	260.3 ± 10.8
EXP	433.2 ± 12.4	492 ± 42.6 ^#^	519.6 ± 73.9 ^#^	535.9 ± 45.3	668.8 ± 73.4 ^#^	701.5 ± 61.7 ^#^
Glucose(mg/dL)						
Start	113.2 ± 8.8	113.8 ± 7.9	122.5 ± 9.3	114.4 ± 7.6	108.1 ± 12.6	105.7 ± 11
EXP	124.7 ± 15.7	197.7 ± 30.1 ^#^	185.6 ± 21.5 ^#^	118.6 ± 21.5	203.5 ± 21.5 ^#^	174 ± 21.4 ^#^
K (mmol/L)	3.3 ± 0.3	3.9 ± 0.5 ^#^	3.51 ± 0.5	3.3 ± 0.2	3.6 ± 0.2	3.5 ± 0.3
Ca (mg/dL)	2.1 ± 0.2	2.2 ± 0.2	2.1 ± 0.2	2.3 ± 0.1	2.3 ± 0.2	2.2 ± 0.1
Cre (mg/dL)	0.28 ± 0.1	0.32 ± 0.1	0.3 ± 0.1	0.35 ± 0.1	0.38 ± 0.07	0.35 ± 0.1
T-Cho (mg/dL)	48.6 ± 7.3	67.1 ± 11.8 ^#^	53.7 ± 9.2 *	59.4 ± 9.9	71.1 ± 12.2	68.2 ± 13
TG (mg/dL)	46 ± 10.3	152.9 ± 65.3 ^#^	93.4 ± 29.9 ^#^	79.5 ± 32	132 ± 46	101 ± 35
HDL (mg/dL)	29.5 ± 5.8	43.8 ± 10.7 ^#^	33.8 ± 8.5	38.8 ± 9.1	47.7 ± 10.3 ^#^	46.2 ± 9.7
LDL (mg/dL)	10.5 ± 1.7	11.5 ± 3.2	13.2 ± 2.8	12.2 ± 2.8	12.9 ± 4	12.9 ± 4
AST (IU/L)	95 ± 43.8	77.4 ± 22.4	69.6 ± 15.7	64.3 ± 11.9	87.9 ± 32.5	75.1 ± 13.5
ALT (IU/L)	36 ± 7.6	53.5 ± 17.5 ^#^	63.8 ± 13.2 ^#^	48.3 ± 13.9	54.3 ± 15.7	67.9 ± 20.9

Values represent mean ± standard deviation. Start, start of the experiment; EXP, explantation; K, potassium; Ca, calcium; Cre; creatinine; T-Cho, total cholesterol; TG, triglycerides; HDL, high-density lipoprotein; LDL, low-density lipoprotein; AST, aspartate aminotransferase; ALT, alanine aminotransferase. ^#^
*p* < 0.05(vs. Control), * *p* < 0.05 (vs. DM).

## Data Availability

The data presented in this study are available on request from the corresponding authors.

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
