# Peer review of "PPAR-Gamma Activation May Inhibit the In Vivo Degeneration of Bioprosthetic Aortic and Aortic Valve Grafts under Diabetic Conditions"

_ijms, 2021, doi:10.3390/ijms222011081_

Round 1

Reviewer 1 Report

In vivo results of an challenging experimental setup to transplant an aortic valve bearing aorta tissue graft using reliable numbers of animals per group are presented in the manuscript from Katahira et al.. Nevertheless rationale and interpretation of datasets has to be fundamentally revised to understand and correctly determine the conclusion.

The following points are unclear:

  • Title does not fit the content of the manuscript because main tissue investigated is the aorta. The results on the AV do not support the statement of the title! Only in the commissure region of AV the Pio-treatment resulted in the fact, that there was no significant higher von Kossa score in the DM-Pio group (tendency??) as observed in the DM group. There is no significant difference between the group DM and DM-Pio.
  • Abstract: change “anti-calcific” and “anti-inflammatory” in the order, as the calcification is the focus of your manuscript. Maybe the inflammation can be part of the supplementary section at al?
  • “Combining this model with preliminary results on a potentially beneficial effect of pioglitazone on prosthetic valve deterioration this work is intended to evaluate the potential of PPARγ activation for improving the chronic functional durability of bio prosthetic valves in diabetic recipients undergoing aortic valve implantation.” This is the aim, as it is written in the end of the introduction. As the results are mainly on grafted aorta tissue, investigation of this tissue should also be mentioned as well as blood parameters. It is a characterization of the entire graft and the animal model instead the AVs only.
  • The authors state, that graft material was generated as described previously. It should be described in the materials and methods section how the aortic valve bearing aortal grafts were generated. There seems to be a decellularization and additional cryoconservation. Please describe the clinical relationship for this preparation. Are there any clinical graft materials that are generated like that and therefore cause the application of this procedure? Are alternatives possible?
  • If there is a decellularization of the graft material, how is the phase “media proliferation” explained. There should be no cells to proliferate. In material and methods section it is written “Donor graft preparation, induction of regurgitation of recipient native aortic 272 valve and heterotopic bioprosthetic aortic valve implantation were performed in a stand-273 ardized fashion as previously reported [21,22] (Supplemental Figure 4).” Literature number 21 includes the decellularization: ….Soon after the harvesting, the aortic graft conduits were decellularized according to a detergent-based protocol consisting of 4 cycles of 12 h incubation with 0.5% sodium dodecyl sulphate, 0.5% sodium deoxycholate and 0.05% (g/v) sodium azide (Sigma-Aldrich, Taufkirchen, Germany), followed by 24 h rinsing with distilled water containing 0.05% sodium azide, and 3 subsequent cycles of 24 h with PBS containing 1% penicillin/streptomycin….”
  • The same question therefore arises for the section 2.4 and Figure 5 showing cells and respective remodelling
  • Authors use a rat model to induce DM and apply pioglitazone to investigate the graft material. Please also discuss or show the results for the aorta and aortic valve of the recipient animal itself. Are there any changes according diabetes and substance treatement?
  • Introduction: first word “recently” for a literature from 2006… Is there more relevant literature correlating diabetes with CAVD?
  • Most effects are observed after 4 weeks, raised adiponectin levels are also observed after 12 weeks – how can this be discussed.
  • Section 2.2.: It is mentioned that aortal graft sections A2, B1 and B2 are used for histological investigation of aorta. Of how many animals and how was evaluation performed exactly?
  • In general: insert number of samples shown in the analyses in the diagrams and mention statistical test. Description of “*” etc.
  • Figure legend 3: improve phrase: “but it was not so different”
  • Insert “4 weeks” and “12 weeks” in Figure 3
  • Why is the media thickness in DM+Pio group after 12 weeks lower that in the control. Does a control group without DM but Pio make sense?
  • How exactly was von Kossa scoring performed? Show examples.
  • Histological Figures show higher magnification of example regions of interest.
  • How was discrimination of leaflet, annulus and commissure performed?
  • Section 2.4. à of how many samples the IF was performed? Why is there no alphaSMA stain at all in the control sample? Are the samples indeed decellularized?
  • New headline with the beginning of line 163? “Inflammatory markers…”
  • Content of Figure 6 can be shown after Figure 4 as the expression of the osteogenic markers is relevant for the aortic calcification.
  • Why is the expression of BMP2 and OPN only different at 4 weeks but not at week 12?
  • Discussion: Data on AV are the smaller part of the results section and data on aorta should be mentioned first. The datasets on calcification differ in their observation and should be discussed carefully. Avoid “we believe”.
  • Discussion: It can be implemented that although AV bioprosthesis calcification but also calcification of aorta graft material can be investigated using this model biomechanics of course differ in the heterotopic position.
  • Limitations: What does this mean?: “More-263 over, due to the limited aortic valve tissue available from each animal, histological analyses and PCR experiments were performed on explants derived from different rats. Hence, histological results are not necessarily identical to the results of gene expression analysis.”
  • Please insert numbers of samples investigated in the different endpoints exactly.
  • “Conclusions” after statistical analysis?
  • Conclusions: “Bioprosthetic aortic valves” does not fit the material investigated.

Author Response

We wish to express our appreciation to the reviewer for his/her insightful comments, which have helped us significantly to improve the quality of our manuscript. We have responded to the comments and added a clinical message.

Reviewer 2 Report

Comments:

  1. Introduction: I would recommend mentioning the background of pioglitazone effects on valvular calcification
  2. Result, first paragraph: table and text should be consistent (both by days or both by weeks).
  3. Result: author should look at the gene expression or makers related to collagen, fibrosis, actin, oxidative stress or other inflammatory markers (RANKL/RANK/OPG)?

Author Response

RESPONSE to Reviewer 2

Comment 1:Introduction: I would recommend mentioning the background of pioglitazone effects on valvular calcification

Answers 1: We agree with your opinion.

Changes 2:We inserted the following sentence in the last paragraph of the introduction.

 “Previously, a translational model for chronic in vivo evaluation of function and durability of biological aortic valvular grafts has been reported to provide valuable reproducible results. Although there are reports that pioglitazone suppresses calcification of the native aortic valve, little is known on its effect on bioprosthetic valves and aortic grafts. By combining this model with a well-established model of diabetes preliminary results on a we sought to evaluate the potentially beneficial effect of pioglitazone on calcific degeneration of bioprosthetic valve bearing aortic grafts. Histological analysis, blood measurements, as well as gene expression analyses were performed. This work is intended to evaluate the potential of PPARγ activation for improving the chronic functional durability of bio prosthetic grafts in diabetic recipients undergoing implantation of aortic grafts.”

-Comment 2:Result, first paragraph: table and text should be consistent (both by days or both by weeks).

Answers 2: We think this suggestion is reasonable. Thank you.

Changes 2: Accordingly, we have modified the text as follows in Table 1.: 28 days to 4 week, 84 days to 12 week.

-Comment 3

Result: author should look at the gene expression or makers related to collagen, fibrosis, actin, oxidative stress or other inflammatory markers (RANKL/RANKIOPG)?

Answer3: We thank you for this suggestion. Of course tissue remodelling is one of the key processes that lead to functional deterioration of bio-prosthetic valves, i.e. the major end point analysed here. In order to follow your suggestions we have performed additional experimental work. Retain cDNA samples of explanted grafts (n=6 animals for each group) were used for gene expression analysis evaluating the following genes:

Osteoprotegerin (OPG), nuclear factor kappa-light-chain-enhancer of activate B-cells (NFkb), monocyte chemoattractant protein 1 (MCP1), matrix metalloproteinases 2 and 9 (MMP2 and MMP9), smooth muscle actin (ACTA2).

We found no regulation of NFkb or MCP1. However, at early time point of 4 weeks we found significantly lower levels of OPG and a trend of lower levels of MMP2 under combined diabetic and pioglitazone treatment when compared to controls. ACTA2 regulation was significantly enhanced at late time point (12 weeks) under pioglitazone treatment.

Change3: We have added the following statement to Discussion: Also changed Supplementary Materials

“Here we observed significantly lower media calcification levels as well as significantly lower gene expression levels for two crucial markers of chondroosteogenic transformation, i.e. osteopontin and bone morphogenic protein, when pioglitazone was administered to diabetic aortic valve recipients. Moreover, other markers of tissue remodeling, i.e. MMP2 and 9 as well as alpha smooth muscle actin were modified in their temporal gene expression during the chronic postoperative period in the DM+Pio group, which may explain the differences in media proliferation and calcification as a result of ours. We speculate that pioglitazone may have acted as an inhibitor on the pathological transformation of vascular smooth muscle cells to osteoblast-like cells. However, our RT-PCR results could not identify inflammatory processes as a possible mechanism by which pioglitazone may have interfered with medial calcification.”

Supplementary Materials: The following are available online at www.mdpi.com/xxx/s1, Figure S1: Alizarin red staining of aortic region, Figure S2: Alizarin red staining of aortic valve region, Figure S3: Gene expression of inflammation markers, Figure S4: Gene expression analysis of targets related to bio-prosthetic valve degeneration. Figure S5: Aortic valve insufficiency operation (A, B), cryopreserved graft (C), Table S1: Primers and respective sequences.

Round 2

Reviewer 1 Report

The authors intensely discussed the questions and points metioned in the review. Unfortunately some aspects were not included in the manuscript materials and methods section or discussion as well. I recommend to add points 6,8,12,19 in the discussion section as well. To add points 5,13,15 and 19 would be helpful for readers to understand the methods and experimental protocols in detail and to evaluate the results critically. At least in part this descriptions can be implemented in the supplementary section.

Author Response

We thank the editorial team and the reviewers for acknowledging our efforts in revising our manuscript. Further we are keen on further improving the manuscript. In the following we would like to highlight our second revision in a point-by-point manner:

Comment 1:

The authors intensely discussed the questions and points metioned in the review. Unfortunately some aspects were not included in the manuscript materials and methods section or discussion as well. I recommend to add points 6,8, 12, 19 in the discussion section as well.

Our response:

Thank you for this suggestion, which we also regard as a valuable extension of our work.

Changes:

  1. We have added point 6 on page 11, line 285 and following.
  2. We have added point 8 on page 12, line 307 and following.
  3. The content of point 12 has been integrated on page 12, lines 327 and following.
  4. Lastly, point 19 has been added to the discussion on page 11, beginning from line 270.

To add points 5, 13, 15 and 19 would be helpful for readers to understand the methods and experimental protocols in detail and to evaluate the results critically. At least in part this descriptions can be implemented in the supplementary section.

Our response:

Thank you for this suggestion. The respective method description have now been focussed in the new Supplemental methods file.

Changes:

Supplemental methods has been added.

For clear presentation purposes, we have now highlighted the changes to the main manuscript. Older Changes from first revision round have now been adapted and only the current revision changes are highlighted.

We thank again the editors and reviewers for the time and effort that they have spent in improving our manuscript.

Payam Akhyari